# The Early Phase of Neural Network Training

**Jonathan Frankle**[†]
MIT CSAIL

**David J. Schwab**
CUNY ITS
Facebook AI Research

**Ari S. Morcos**
Facebook AI Research

## Abstract

Recent studies have shown that many important aspects of neural network learning take place within the very earliest iterations or epochs of training. For example, sparse, trainable sub-networks emerge (Frankle et al., 2019), gradient descent moves into a small subspace (Gur-Ari et al., 2018), and the network undergoes a critical period (Achille et al., 2019). Here we examine the changes that deep neural networks undergo during this early phase of training. We perform extensive measurements of the network state during these early iterations of training and leverage the framework of Frankle et al. (2019) to quantitatively probe the weight distribution and its reliance on various aspects of the dataset. We find that, within this framework, deep networks are not robust to reinitializing with random weights while maintaining signs, and that weight distributions are highly non-independent even after only a few hundred iterations. Despite this behavior, pre-training with blurred inputs or an auxiliary self-supervised task can approximate the changes in supervised networks, suggesting that these changes are not inherently label-dependent, though labels significantly accelerate this process. Together, these results help to elucidate the network changes occurring during this pivotal initial period of learning.

## 1 Introduction

Over the past decade, methods for successfully training big, deep neural networks have revolutionized machine learning. Yet surprisingly, the underlying reasons for the success of these approaches remain poorly understood, despite remarkable empirical performance (Santurkar et al., 2018; Zhang et al., 2017). A large body of work has focused on understanding what happens during the later stages of training (Neyshabur et al., 2019; Yaida, 2019; Chaudhuri & Soatto, 2017; Wei & Schwab, 2019), while the initial phase has been less explored. However, a number of distinct observations indicate that significant and consequential changes are occurring during the most early stage of training. These include the presence of critical periods during training (Achille et al., 2019), the dramatic reshaping of the local loss landscape (Sagun et al., 2017; Gur-Ari et al., 2018), and the necessity of rewinding in the context of the lottery ticket hypothesis (Frankle et al., 2019). Here we perform a thorough investigation of the state of the network in this early stage.

To provide a unified framework for understanding the changes the network undergoes during the early phase, we employ the methodology of iterative magnitude pruning with rewinding (IMP), as detailed below, throughout the bulk of this work (Frankle & Carbin, 2019; Frankle et al., 2019). The initial lottery ticket hypothesis, which was validated on comparatively small networks, proposed that small, sparse sub-networks found via pruning of converged larger models could be trained to high performance provided they were initialized with the same values used in the training of the unpruned model (Frankle & Carbin, 2019). However, follow-up work found that rewinding the weights to their values at some iteration early in the training of the unpruned model, rather than to their initial values, was necessary to achieve good performance on deeper networks such as ResNets (Frankle et al., 2019). This observation suggests that the changes in the network during this initial phase are vital for the success of the training of small, sparse sub-networks. As a result, this paradigm provides a simple and quantitative scheme for measuring the importance of the weights at various points early in training within an actionable and causal framework.

---

[†]Work done while an intern at Facebook AI Research.

We make the following contributions, all evaluated across three different network architectures:

1. We provide an in-depth overview of various statistics summarizing learning over the early part of training.

2. We evaluate the impact of perturbing the state of the network in various ways during the early phase of training, finding that:

    (i) counter to observations in smaller networks (Zhou et al., 2019), deeper networks are not robust to reinitializion with random weights, but maintained signs

    (ii) the distribution of weights after the early phase of training is already highly non-i.i.d., as permuting them dramatically harms performance, even when signs are maintained

    (iii) both of the above perturbations can roughly be approximated by simply adding noise to the network weights, though this effect is stronger for (ii) than (i)

3. We measure the data-dependence of the early phase of training, finding that pre-training using only $p(x)$ can approximate the changes that occur in the early phase of training, though pre-training must last for far longer ($\sim 32\times$ longer) and not be fed misleading labels.

## 2    Known Phenomena in the Early Phase of Training

**Lottery ticket rewinding:** The original lottery ticket paper (Frankle & Carbin, 2019) rewound weights to initialization, i.e., $k = 0$, during IMP. Follow up work on larger models demonstrated that it is necessary to rewind to a later point during training for IMP to succeed, i.e., $k << T$, where $T$ is total training iterations (Frankle et al., 2019). Notably, the benefit of rewinding to a later point in training saturates quickly, roughly between 500 and 2000 iterations for ResNet-20 on CIFAR-10 (Figure 1). This timescale is strikingly similar to the changes in the Hessian described below.

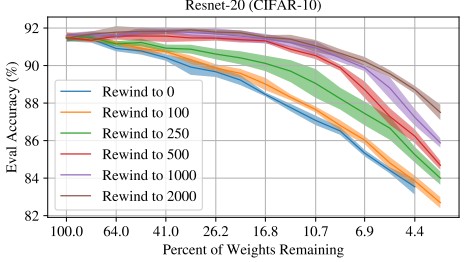

Figure 1: Accuracy of IMP when rewinding to various iterations of the early phase for ResNet-20 sub-networks as a function of sparsity level.

**Hessian eigenspectrum:** The shape of the loss landscape around the network state also appears to change rapidly during the early phase of training (Sagun et al., 2017; Gur-Ari et al., 2018). At initialization, the Hessian of the loss contains a number of large positive and negative eigenvalues. However, very rapidly the curvature is reshaped in a few marked ways: a few large eigenvalues emerge, the bulk eigenvalues are close to zero, and the negative eigenvalues become very small. Moreover, once the Hessian spectrum has reshaped, gradient descent appears to occur largely within the top subspace of the Hessian (Gur-Ari et al., 2018). These results have been largely confirmed in large scale studies (Ghorbani et al., 2019), but note they depend to some extent on architecture and (absence of) batch normalization (Ioffe & Szegedy, 2015). A notable exception to this consistency is the presence of substantial $L_1$ energy of negative eigenvalues for models trained on ImageNet.

**Critical periods in deep learning:** Achille et al. (2019) found that perturbing the training process by providing corrupted data early on in training can result in irrevocable damage to the final performance of the network. Note that the timescales over which the authors find a critical period extend well beyond those we study here. However, architecture, learning rate schedule, and regularization all modify the timing of the critical period, and follow-up work found that critical periods were also present for regularization, in particular weight decay and data augmentation (Golatkar et al., 2019).

## 3    Preliminaries and Methodology

**Networks:** Throughout this paper, we study five standard convolutional neural networks for CIFAR-10. These include the ResNet-20 and ResNet-56 architectures designed for CIFAR-10 (He et al., 2015), the ResNet-18 architecture designed for ImageNet but commonly used on CIFAR-10 (He et al., 2015), the WRN-16-8 wide residual network (Zagoruyko & Komodakis, 2016), and the

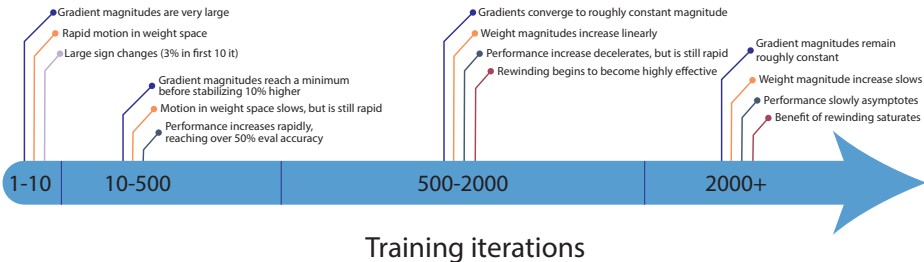

Figure 2: Rough timeline of the early phase of training for ResNet-20 on CIFAR-10.

VGG-13 network (Simonyan & Zisserman (2015) as adapted by Liu et al. (2019)). Throughout the main body of the paper, we show ResNet-20; in Appendix B, we present the same experiments for the other networks. Unless otherwise stated, results were qualitatively similar across all three networks. All experiments in this paper display the mean and standard deviation across five replicates with different random seeds. See Appendix A for further model details.

**Iterative magnitude pruning with rewinding:** In order to test the effect of various hypotheses about the state of sparse networks early in training, we use the *Iterative Magnitude Pruning with rewinding* (IMP) procedure of Frankle et al. (2019) to extract sub-networks from various points in training that could have learned on their own. The procedure involves training a network to completion, pruning the 20% of weights with the lowest magnitudes globally throughout the network, and *rewinding* the remaining weights to their values from an earlier iteration $k$ during the initial, pre-pruning training run. This process is iterated to produce networks with high sparsity levels. As demonstrated in Frankle et al. (2019), IMP with rewinding leads to sparse sub-networks which can train to high performance even at high sparsity levels $> 90\%$.

Figure 1 shows the results of the IMP with rewinding procedure, showing the accuracy of ResNet-20 at increasing sparsity when performing this procedure for several rewinding values of $k$. For $k \geq 500$, sub-networks can match the performance of the original network with 16.8% of weights remaining. For $k > 2000$, essentially no further improvement is observed (not shown).

## 4    THE STATE OF THE NETWORK EARLY IN TRAINING

Many of the aforementioned papers refer to various points in the "early" part of training. In this section, we descriptively chart the state of ResNet-20 during the earliest phase of training to provide context for this related work and our subsequent experiments. We specifically focus on the first 4,000 iterations (10 epochs). See Figure A3 for the characterization of additional networks. We include a summary of these results for ResNet-20 as a timeline in Figure 2, and include a broader timeline including results from several previous papers for ResNet-18 in Figure A1.

As shown in Figure 3, during the earliest ten iterations, the network undergoes substantial change. It experiences large gradients that correspond to a rapid increase in distance from the initialization and a large number of sign changes of the weights. After these initial iterations, gradient magnitudes drop and the rate of change in each of the aforementioned quantities gradually slows through the remainder of the period we observe. Interestingly, gradient magnitudes reach a minimum after the first 200 iterations and subsequently increase to a stable level by iteration 500. Evaluation accuracy, improves rapidly, reaching 55% by the end of the first epoch (400 iterations), more than halfway to the final 91.5%. By 2000 iterations, accuracy approaches 80%.

During the first 4000 iterations of training, we observe three sub-phases. In the first phase, lasting only the initial few iterations, gradient magnitudes are very large and, consequently, the network changes rapidly. In the second phase, lasting about 500 iterations, performance quickly improves, weight magnitudes quickly increase, sign differences from initialization quickly increase, and gradient magnitudes reach a minimum before settling at a stable level. Finally, in the third phase, all of these quantities continue to change in the same direction, but begin to decelerate.

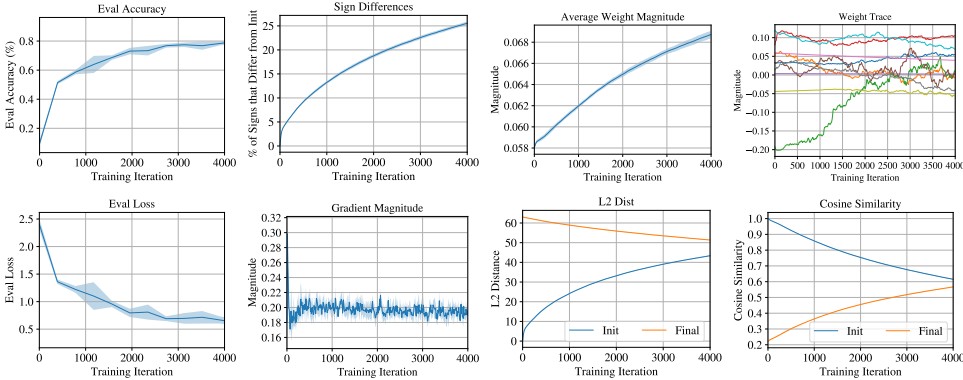

Figure 3: Basic telemetry about the state of ResNet-20 during the first 4000 iterations (10 epochs). Top row: evaluation accuracy/loss; average weight magnitude; percentage of weights that change sign from initialization; the values of ten randomly-selected weights. Bottom row: gradient magnitude; L2 distance of weights from their initial values and final values at the end of training; cosine similarity of weights from their initial values and final values at the end of training.

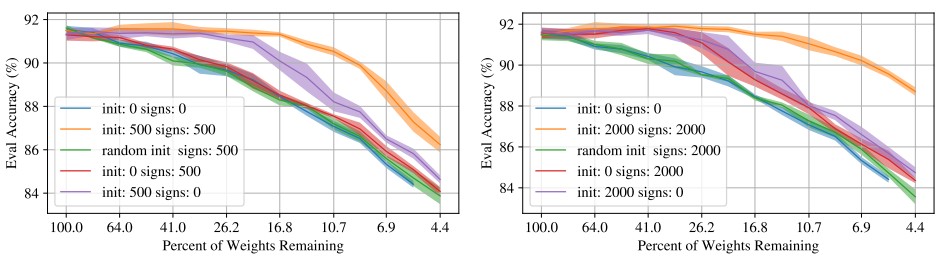

Figure 4: Performance of an IMP-derived sub-network of ResNet-20 on CIFAR-10 initialized to the signs at iteration 0 or $k$ and the magnitudes at iteration 0 or $k$. Left: $k = 500$. Right: $k = 2000$.

## 5 PERTURBING NEURAL NETWORKS EARLY IN TRAINING

Figure 1 shows that the changes in the network weights over the first 500 iterations of training are essential to enable high performance at high sparsity levels. What features of this weight transformation are necessary to recover increased performance? Can they be summarized by maintaining the weight signs, but discarding their magnitudes as implied by Zhou et al. (2019)? Can they be represented distributionally? In this section, we evaluate these questions by perturbing the early state of the network in various ways. Concretely, we either add noise or shuffle the weights of IMP sub-networks of ResNet-20 across different network sub-compenents and examine the effect on the network's ability to learn thereafter. The sub-networks derived by IMP with rewinding make it possible to understand the causal impact of perturbations on sub-networks that are as capable as the full networks but more visibly decline in performance when improperly configured. To enable comparisons between the experiments in Section 5 and provide a common frame of reference, we measure the effective standard deviation of each perturbation, i.e. stddev($w_{perturb} - w_{orig}$).

### 5.1 ARE SIGNS ALL YOU NEED?

Zhou et al. (2019) show that, for a set of small convolutional networks, signs alone are sufficient to capture the state of lottery ticket sub-networks. However, it is unclear whether signs are still sufficient for larger networks early in training. In Figure 4, we investigate the impact of combining the magnitudes of the weights from one time-point with the signs from another. We found that the signs at iteration 500 paired with the magnitudes from initialization (red line) or from a separate random initialization (green line) were insufficient to maintain the performance reached by using both signs and magnitudes from iteration 500 (orange line), and performance drops to that of using

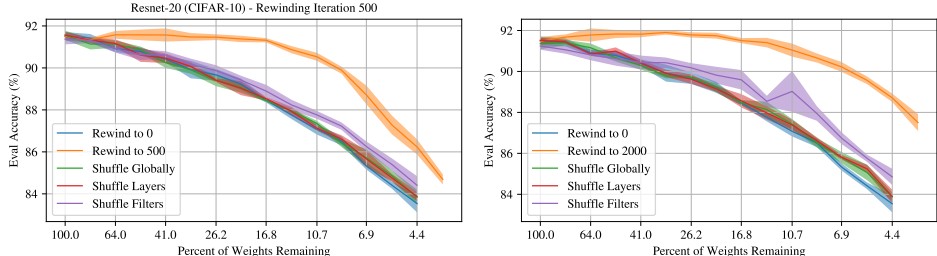

Figure 5: Performance of an IMP-derived ResNet-20 sub-network on CIFAR-10 initialized with the weights at iteration $k$ permuted within various structural elements. Left: $k = 500$. Right: $k = 2000$.

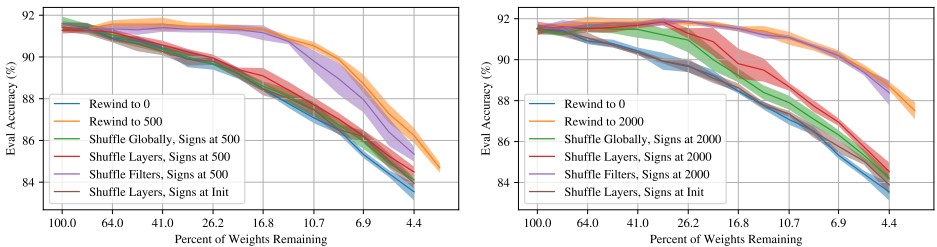

Figure 6: The effect of training an IMP-derived sub-network of ResNet-20 on CIFAR-10 initialized with the weights at iteration $k$ as shuffled within various structural elements where shuffling only occurs between weights with the same sign. Left: $k = 500$. Right: $k = 2000$.

both magnitudes and signs from initialization (blue line). However, while using the magnitudes from iteration 500 and the signs from initialization, performance is still substantially better than initialization signs and magnitudes. In addition, the overall perturbation to the network by using the magnitudes at iteration 500 and signs from initialization (mean: 0.0, stddev: 0.033) is smaller than by using the signs at iteration 500 and the magnitudes from initialization ($0.0 \pm 0.042$, mean $\pm$ std). These results suggest that the change in weight magnitudes over the first 500 iterations of training are substantially more important than the change in the signs for enabling subsequent training.

By iteration 2000, however, pairing the iteration 2000 signs with magnitudes from initialization (red line) reaches similar performance to using the signs from initialization and the magnitudes from iteration 2000 (purple line) though not as high performance as using both from iteration 2000. This result suggests that network signs undergo important changes between iterations 500 and 2000, as only 9% of signs change during this period. Our results also suggest that counter to the observations of Zhou et al. (2019) in shallow networks, signs are not sufficient in deeper networks.

## 5.2 ARE WEIGHT DISTRIBUTIONS I.I.D.?

Can the changes in weights over the first $k$ iterations be approximated distributionally? To measure this, we permuted the weights at iteration $k$ within various structural sub-components of the network (globally, within layers, and within convolutional filters). If networks are robust to these permutations, it would suggest that the weights in such sub-compenents might be approximated and sampled from. As Figure 5 shows, however, we found that performance was not robust to shuffling weights globally (green line) or within layers (red line), and drops substantially to no better than that of the original initialization (blue line) at both 500 and 2000 iterations.[1] Shuffling within filters (purple line) performs slightly better, but results in a smaller overall perturbation ($0.0 \pm 0.092$ for $k = 500$) than shuffling layerwise ($0.0 \pm 0.143$) or globally ($0.0 \pm 0.144$), suggesting that this change in perturbation strength may simply account for the difference.

---

[1] We also considered shuffling within the incoming and outgoing weights for each neuron, but performance was equivalent to shuffling within layers. We elided these lines for readability.

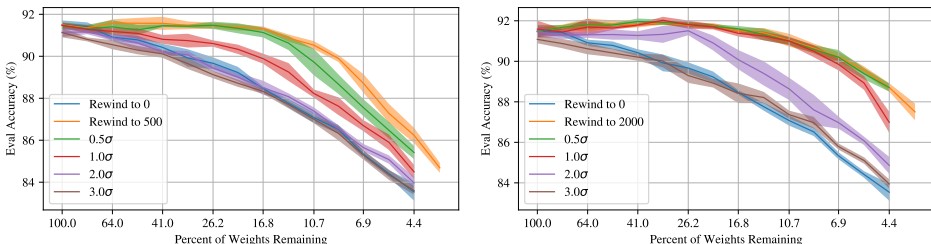

Figure 7: The effect of training an IMP-derived sub-network of ResNet-20 on CIFAR-10 initialized with the weights at iteration $k$ and Gaussian noise of $n\sigma$, where $\sigma$ is the standard deviation of the initialization distribution for each layer. Left: $k = 500$. Right: $k = 2000$.

Are the signs from the rewinding iteration, $k$, sufficient to recover the damage caused by permutation? In Figure 6, we also consider shuffling only amongst weights that have the same sign. Doing so substantially improves the performance of the filter-wise shuffle; however, it also reduces the extent of the overall perturbation ($0.0 \pm 0.049$ for $k = 500$). It also improves the performance of shuffling within layers slightly for $k = 500$ and substantially for $k = 2000$. We attribute the behavior for $k = 2000$ to the signs just as in Figure 4: when the magnitudes are similar in value (Figure 4 red line) or distribution (Figure 6 red and green lines), using the signs improves performance. Reverting back to the initial signs while shuffling magnitudes within layers (brown line), however, damages the network too severely ($0.0 \pm 0.087$ for $k = 500$) to yield any performance improvement over random noise. These results suggest that, while the signs from initialization are not sufficient for high performance at high sparsity as shown in Section 5.1, the signs from the rewinding iteration *are* sufficient to recover the damage caused by permutation, at least to some extent.

## 5.3 Is it all just noise?

Some of our previous results suggested that the impact of signs and permutations may simply reduce to adding noise to the weights. To evaluate this hypothesis, we next study the effect of simply adding Gaussian noise to the network weights at iteration $k$. To add noise appropriately for layers with different scales, the standard deviation of the noise added for each layer was normalized to a multiple of the standard deviation $\sigma$ of the initialization distribution for that layer. In Figure 7, we see that for iteration $k = 500$, sub-networks can tolerate $0.5\sigma$ to $1\sigma$ of noise before performance degrades back to that of the original initialization at higher levels of noise. For iteration $k = 2000$, networks are surprisingly robust to noise up to $1\sigma$, and even $2\sigma$ exhibits nontrivial performance.

In Figure 8, we plot the performance of each network at a fixed sparsity level as a function of the effective standard deviation of the noise imposed by each of the aforementioned perturbations. We find that the standard deviation of the effective noise explained fairly well the resultant performance ($k = 500$: $r = -0.672$, $p = 0.008$; $k = 2000$: $r = -0.726$, $p = 0.003$). As expected, perturbations that preserved the performance of the network generally resulted in smaller changes to the state of the network at iteration $k$. Interestingly, experiments that mixed signs and magnitudes from different points in training (green points) aligned least well with this pattern: the standard deviation of the perturbation is roughly similar among all of these experiments, but the accuracy of the resulting networks changes substantially. This result suggests that although the standard deviation of the noise is certainly indicative of lower accuracy, there are still specific perturbations that, while small in overall magnitude, can have a large effect on the network's ability to learn, suggesting that the observed perturbation effects are not, in fact, just a consequence of noise.

## 6 The Data-Dependence of Neural Networks Early in Training

Section 5 suggests that the change in network behavior by iteration $k$ is not due to easily-ascertainable, distributional properties of the network weights and signs. Rather, it appears that training is required to reach these network states. It is unclear, however, the extent to which various aspects of the data distribution are necessary. Mainly, is the change in weights during the early phase of training dependent on $p(x)$ or $p(y|x)$? Here, we attempt to answer this question by mea-

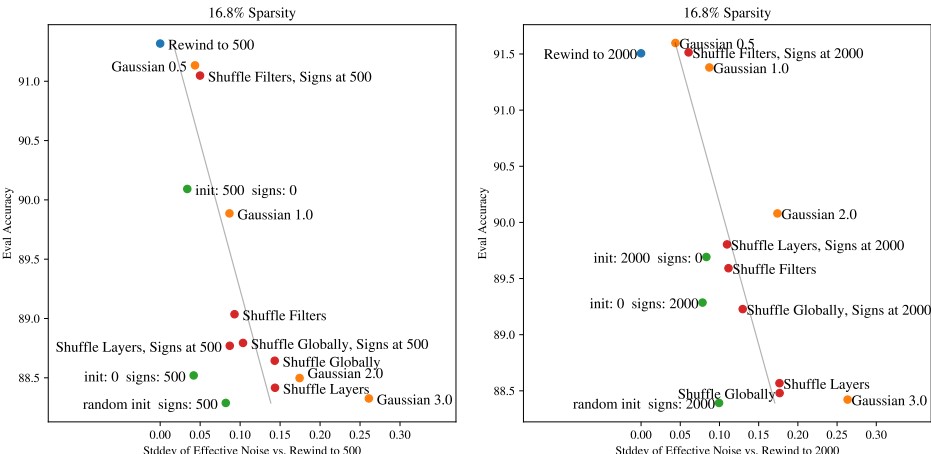

Figure 8: The effective standard deviation of various perturbations as a function of mean evaluation accuracy (across 5 seeds) at sparsity 26.2%. The mean of each perturbation was approximately 0. Left: $k = 500$, $r = -0.672$, $p = 0.008$; Right: $k = 2000$, $r = -0.726$, $p = 0.003$.

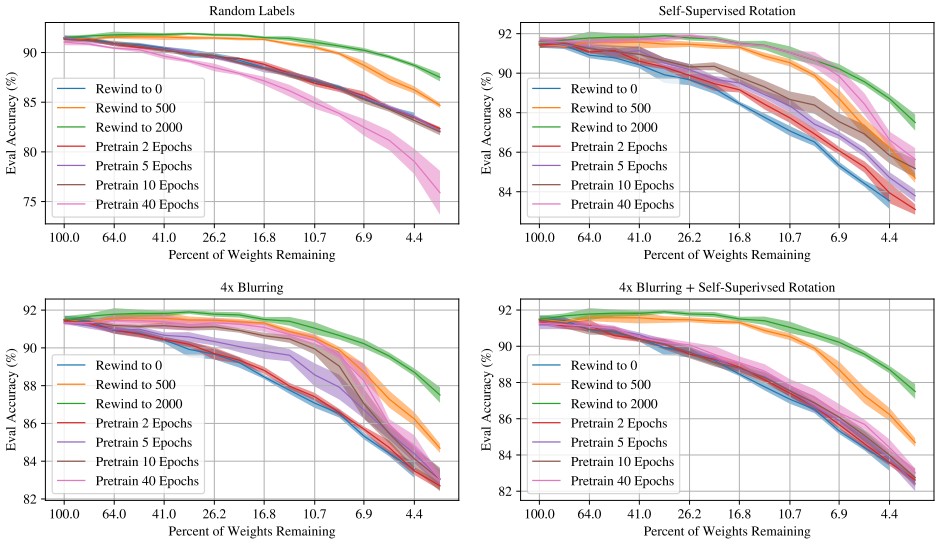

Figure 9: The effect of pre-training ResNet-20 on CIFAR-10 with random labels, self-supervised rotation, 4x blurring, and 4x blurring and self-supervised rotation.

suring the extent to which we can re-create a favorable network state for sub-network training using restricted information from the training data and labels. In particular, we consider pre-training the network with techniques that ignore labels entirely (self-supervised rotation prediction, Section 6.2), provide misleading labels (training with random labels, Section 6.1), or eliminate information from examples (blurring training examples Section 6.3).

We first train a randomly-initialized, unpruned network on CIFAR-10 on the pre-training task for a set number of epochs. After pre-training, we train the network normally as if the pre-trained state were the original initialization. We then use the state of the network at the end of the pre-training phase as the "initialization" to find masks for IMP. Finally, we examine the performance of the IMP-pruned sub-networks as initialized using the state after pre-training. This experiment determines the extent to which pre-training places the network in a state suitable for sub-network training as compared to using the state of the network at iteration $k$ of training on the original task.

## 6.1 RANDOM LABELS

To evaluate whether this phase of training is dependent on underlying structure in the data, we drew inspiration from Zhang et al. (2017) and pre-trained networks on data with randomized labels. This experiment tests whether the input distribution of the training data is sufficient to put the network in a position from which IMP with rewinding can find a sparse, trainable sub-network despite the presence of incorrect (not just missing) labels. Figure 9 (upper left) shows that pre-training on random labels for up to 10 epochs provides no improvement above rewinding to iteration 0 and that pre-training for longer begins to hurt accuracy. This result suggests that, though it is still possible that labels may not be required for learning, the presence incorrect labels is sufficient to prevent learning which approximates the early phase of training.

## 6.2 SELF-SUPERVISED ROTATION PREDICTION

What if we remove labels entirely? Is $p(x)$ sufficient to approximate the early phase of training? Historically, neural network training often involved two steps: a self-supervised pre-training phase followed by a supervised phase on the target task (Erhan et al., 2010). Here, we consider one such self-supervised technique: rotation prediction (Gidaris et al., 2018). During the pre-training phase, the network is presented with a training image that has randomly been rotated $90n$ degrees (where $n \in \{0, 1, 2, 3\}$). The network must classify examples by the value of $n$. If self-supervised pre-training can approximate the early phase of training, it would suggest that $p(x)$ is sufficient on its own. Indeed, as shown in Figure 9 (upper right), this pre-training regime leads to well-trainable sub-networks, though networks must be trained for many more epochs compared to supervised training (40 compared to 1.25, or a factor of $32\times$). This result suggests that the labels for the ultimate task themselves are not necessary to put the network in such a state (although explicitly misleading labels are detrimental). We emphasize that the duration of the pre-training phase required is an order of magnitude larger than the original rewinding iteration, however, suggesting that labels add important information which accelerates the learning process.

## 6.3 BLURRING TRAINING EXAMPLES

To probe the importance of $p(x)$ for the early phase of training, we study the extent to which the training input distribution is necessary. Namely, we pretrain using blurred training inputs with the correct labels. Following Achille et al. (2019), we blur training inputs by downsampling by 4x and then upsampling back to the full size. Figure 9 (bottom left) shows that this pre-training method succeeds: after 40 epochs of pre-training, IMP with rewinding can find sub-networks that are similar in performance to those found after training on the original task for 500 iterations (1.25 epochs).

Due to the success of the the rotation and blurring pre-training tasks, we explored the effect of combining these pre-training techniques. Doing so tests the extent to which we can discard both the training labels and some information from the training inputs. Figure 9 (bottom right) shows that doing so provides the network too little information: no amount of pre-training we considered makes it possible for IMP with rewinding to find sub-networks that perform tangibly better than rewinding to iteration 0. Interestingly however, as shown in Appendix B, trainable sub-networks *are found* for VGG-13 with this pre-training regime, suggesting that different network architectures have different sensitivities to the deprivation of labels and input content.

## 6.4 SPARSE PRETRAINING

Since sparse sub-networks are often challenging to train from scratch without the proper initialization (Han et al., 2015; Liu et al., 2019; Frankle & Carbin, 2019), does pre-training make it easier for sparse neural networks to learn? Doing so would serve as a rough form of curriculum learning (Bengio et al., 2009) for sparse neural networks. We experimented with training sparse sub-networks of ResNet-20 (IMP sub-networks, randomly reinitialized sub-networks, and randomly pruned sub-networks) first on self-supervised rotation and then on the main task, but found no benefit beyond rewinding to iteration 0 (Figure 10). Moreover, doing so when starting from a sub-network rewound to iteration 500 actually hurts final accuracy. This result suggests that while pre-training *is sufficient* to approximate the early phase of supervised training with an appropriately structured mask, it *is not sufficient* to do so with an inappropriate mask.

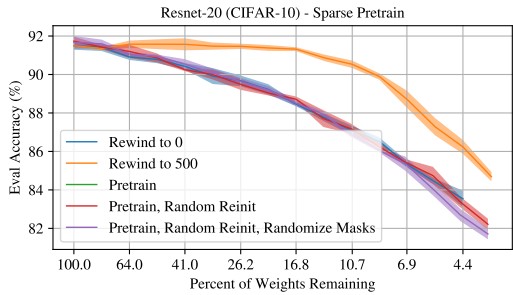

Figure 10: The effect of pretraining sparse sub-networks of Resnet-20 (rewound to iteration 500) with 40 epochs of self-supervised rotation before training on CIFAR-10.

## 7 DISCUSSION

In this paper, we first performed extensive measurements of various statistics summarizing learning over the early part of training. Notably, we uncovered 3 sub-phases: in the very first iterations, gradient magnitudes are anomalously large and motion is rapid. Subsequently, gradients overshoot to smaller magnitudes before leveling off while performance increases rapidly. Then, learning slowly begins to decelerate. We then studied a suite of perturbations to the network state in the early phase finding that, counter to observations in smaller networks (Zhou et al., 2019), deeper networks are not robust to reinitializing with random weights with maintained signs. We also found that the weight distribution after the early phase of training is highly non-independent. Finally, we measured the data-dependence of the early phase with the surprising result that pre-training on a self-supervised task yields equivalent performance to late rewinding with IMP.

These results have significant implications for the lottery ticket hypothesis. The seeming necessity of late rewinding calls into question certain interpretations of lottery tickets as well as the ability to identify sub-networks at initialization. Our observation that weights are highly non-independent at the rewinding point suggests that the weights at this point cannot be easily approximated, making approaches which attempt to "jump" directly to the rewinding point unlikely to succeed. However, our result that labels are not necessary to approximate the rewinding point suggests that the learning during this phase does not require task-specific information, suggesting that rewinding may not be necessary if networks are pre-trained appropriately.

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

# A    MODEL DETAILS

| Network | Epochs | Batch Size | Learning Rate | Parameters | Eval Accuracy |
|---------|--------|-----------|---------------|-----------|---------------|
| ResNet-20 | 160 | 128 | 0.1 (Mom 0.9) | 272K | $91.5 \pm 0.2\%$ |
| ResNet-56 | 160 | 128 | 0.1 (Mom 0.9) | 856K | $93.0 \pm 0.1\%$ |
| ResNet-18 | 160 | 128 | 0.1 (Mom 0.9) | 11.2M | $86.8 \pm 0.3\%$ |
| VGG-13 | 160 | 64 | 0.1 (Mom 0.9) | 9.42M | $93.5 \pm 0.1\%$ |
| WRN-16-8 | 160 | 128 | 0.1 (Mom 0.9) | 11.1M | $94.8 \pm 0.1\%$ |

Table A1: Summary of the networks we study in this paper. We present ResNet-20 in the main body of the paper and the remaining networks in Appendix B.

Table A1 summarizes the networks. All networks follow the same training regime: we train with SGD for 160 epochs starting at learning rate 0.1 (momentum 0.9) and drop the learning rate by a factor of ten at epoch 80 and again at epoch 120. Training includes weight decay with weight 1e-4. Data is augmented with normalization, random flips, and random crops up to four pixels in any direction.

# B    EXPERIMENTS FOR OTHER NETWORKS

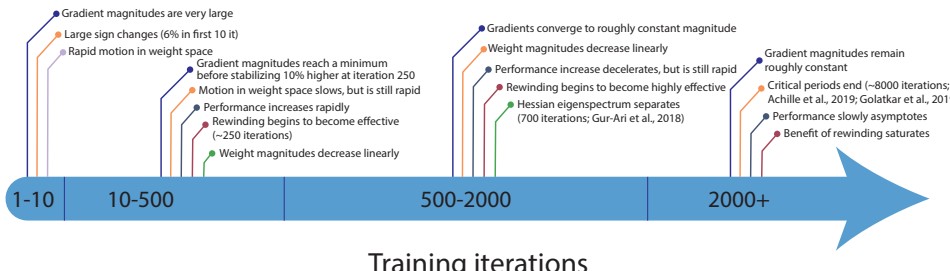

Figure A1: Rough timeline of the early phase of training for ResNet-18 on CIFAR-10, including results from previous papers.

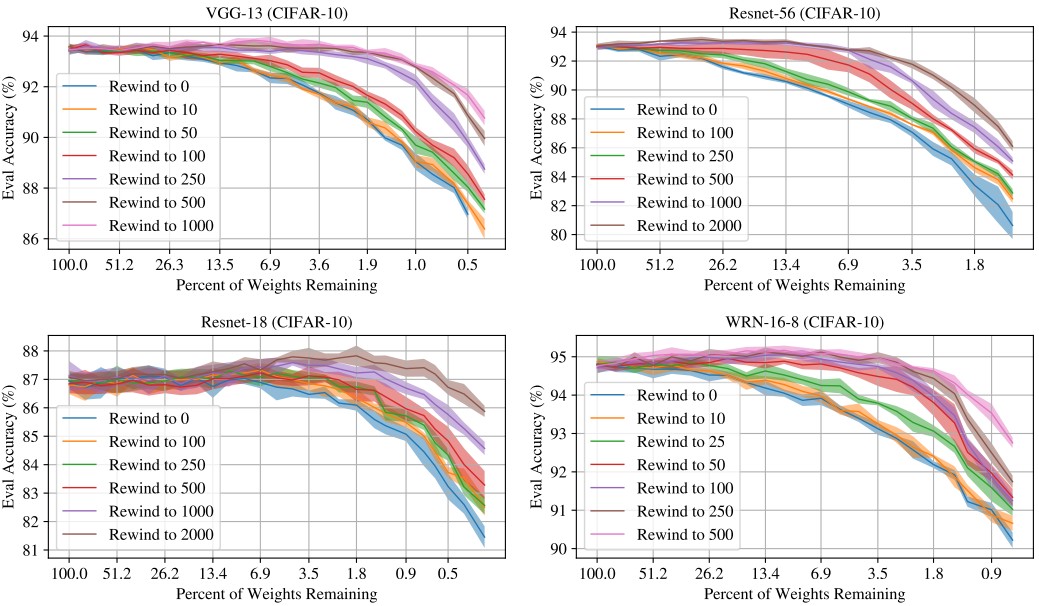

Figure A2: The effect of IMP rewinding iteration on the accuracy of sub-networks at various levels of sparsity. Accompanies Figure 1.

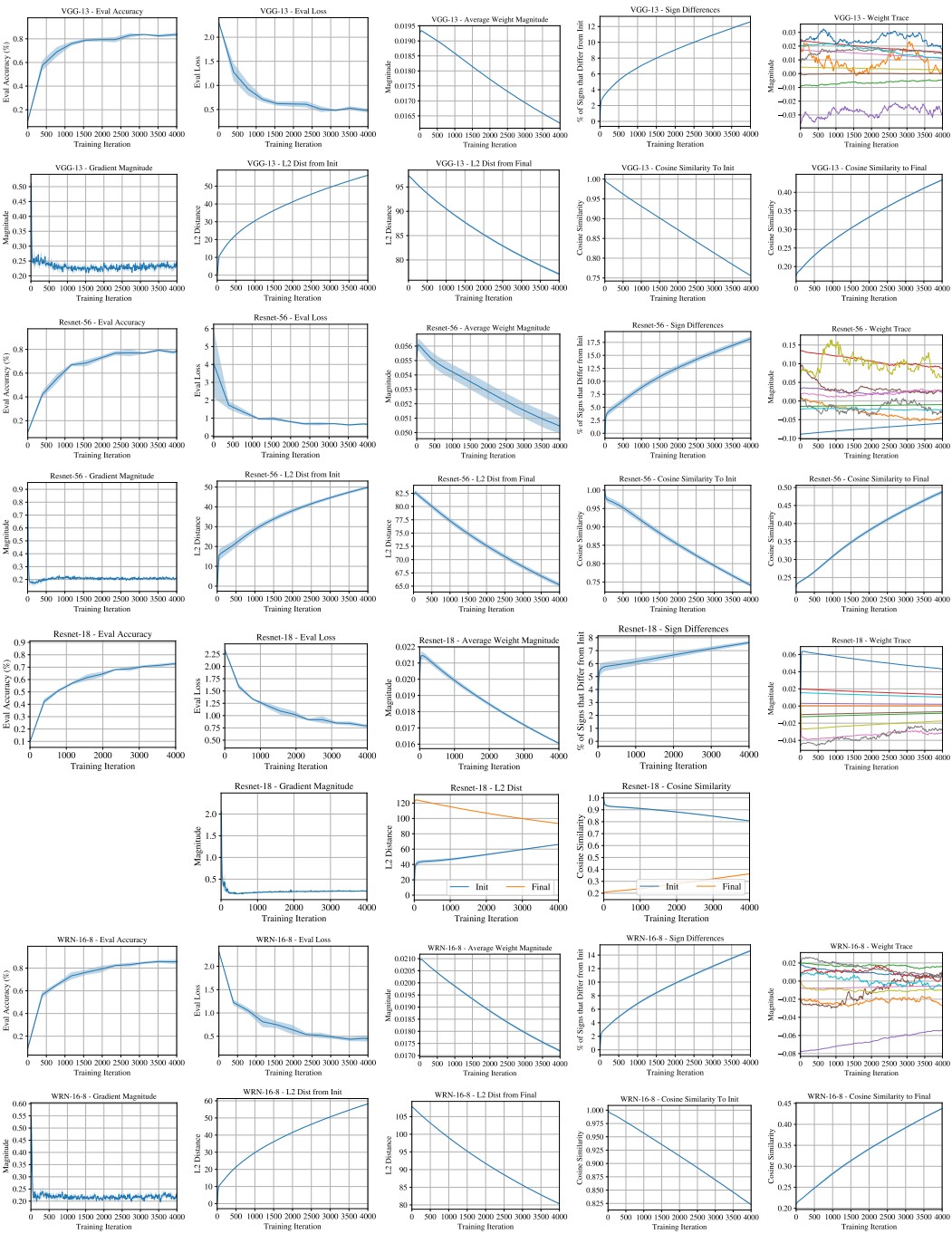

Figure A3: Basic telemetry about the state of all networks in Table A1 during the first 4000 iterations of training. Accompanies Figure 3.

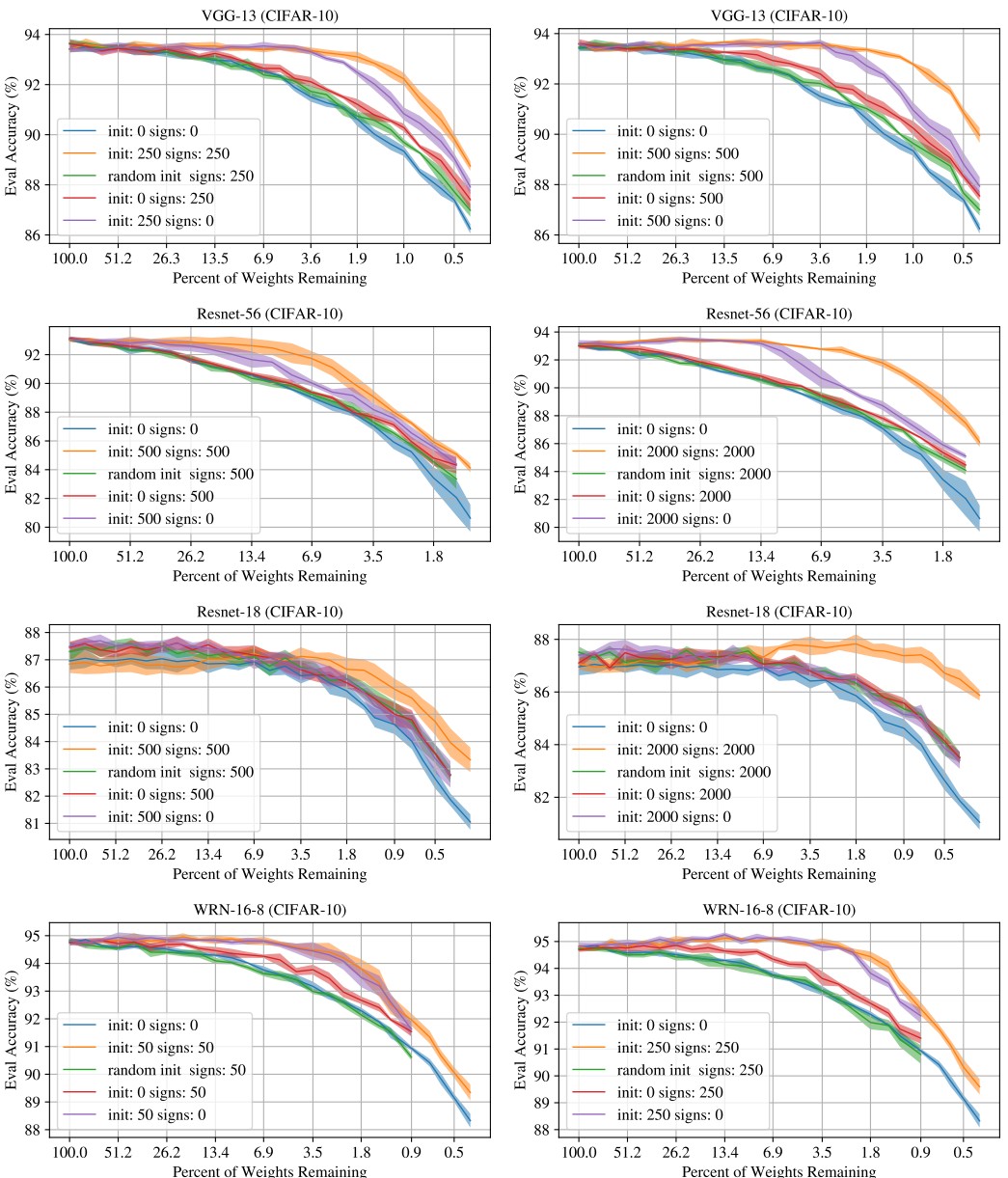

Figure A4: The effect of training an IMP-derived sub-network initialized to the signs at iteration 0 or $k$ and the magnitudes at iteration 0 or $k$. Accompanies Figure A4.

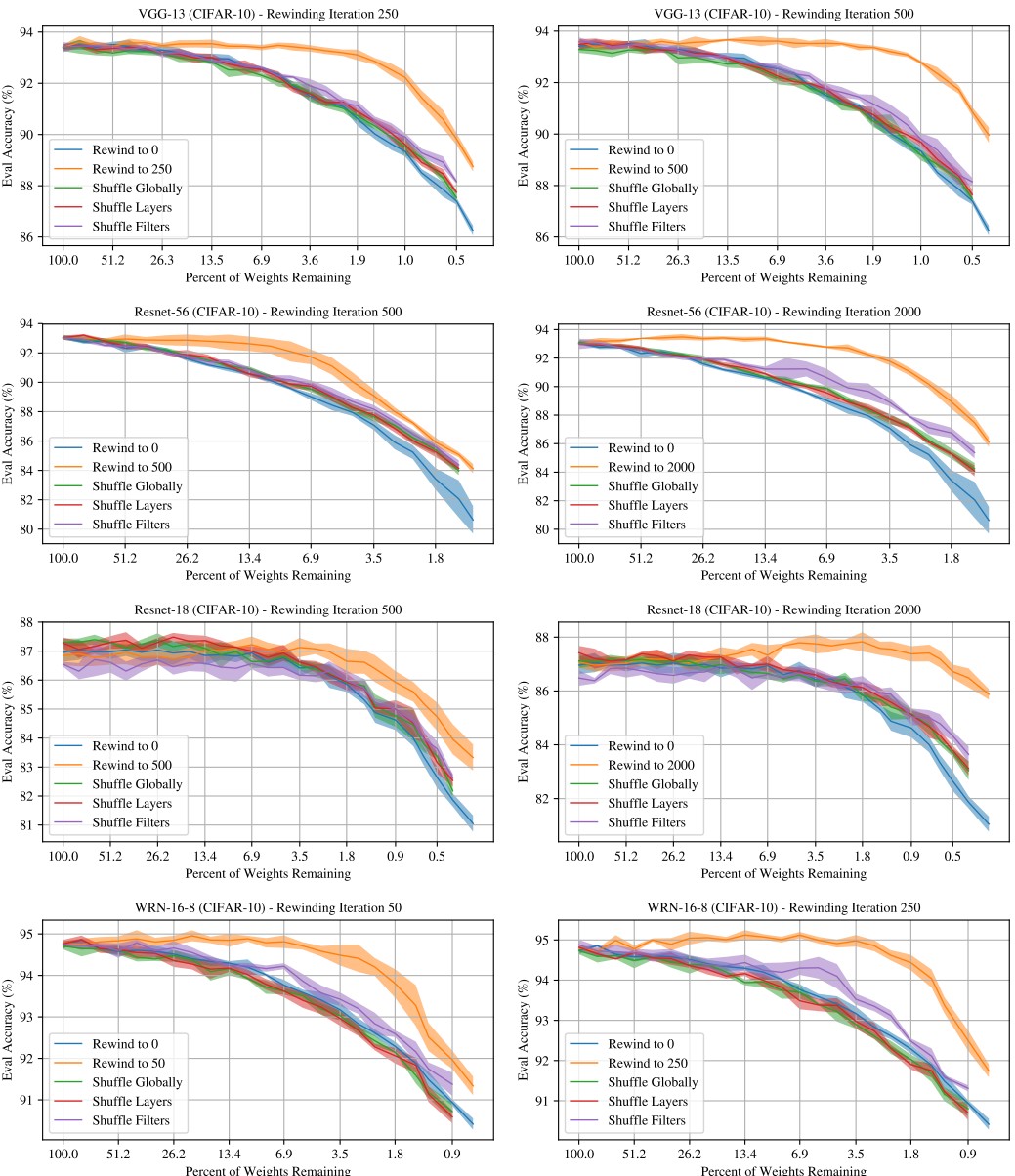

Figure A5: The effect of training an IMP-derived sub-network initialized with the weights at iteration $k$ as shuffled within various structural elements. Accompanies Figure 5.

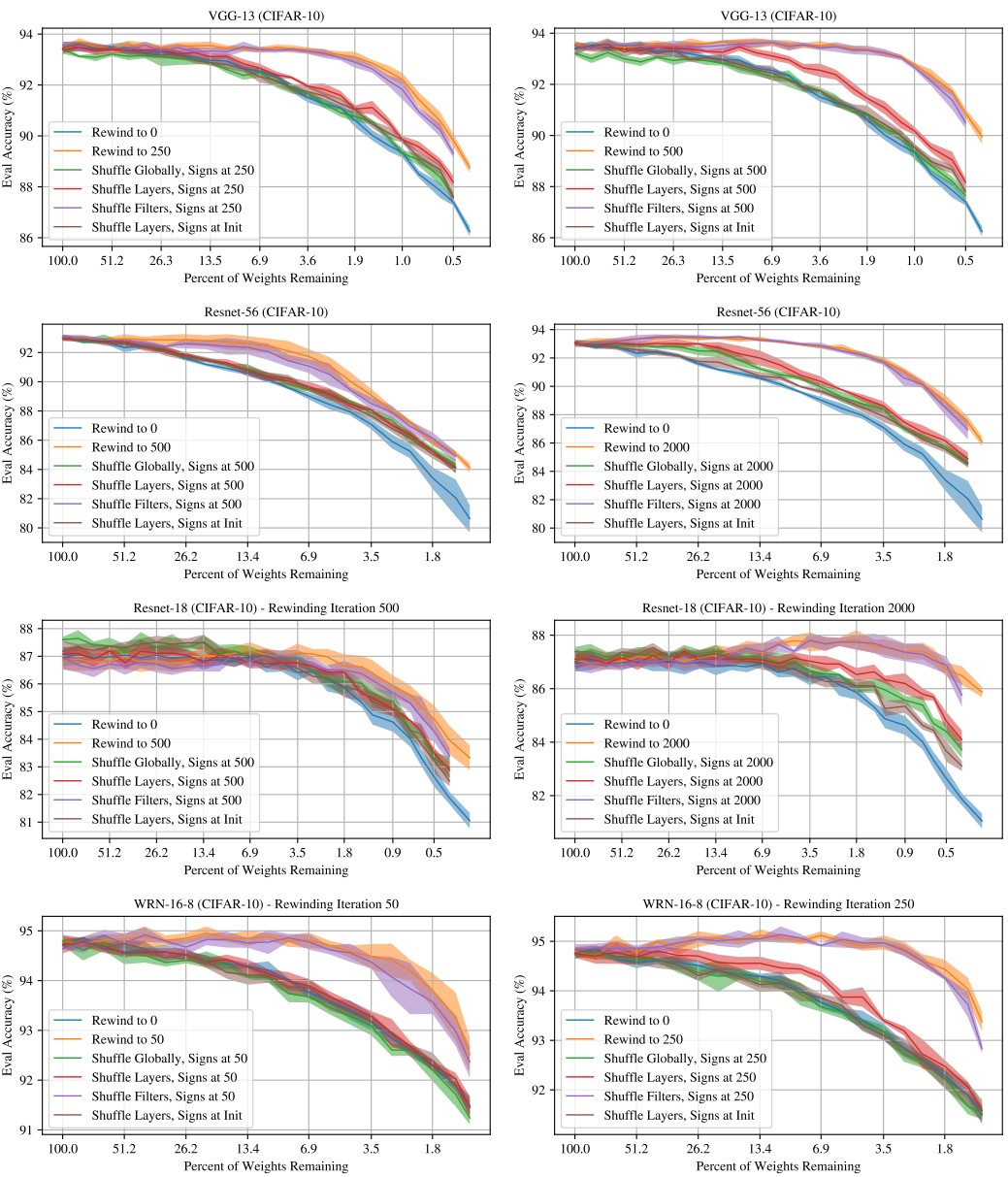

Figure A6: The effect of training an IMP-derived sub-network initialized with the weights at iteration $k$ as shuffled within various structural elements where shuffling only occurs between weights with the same sign. Accompanies Figure 6.

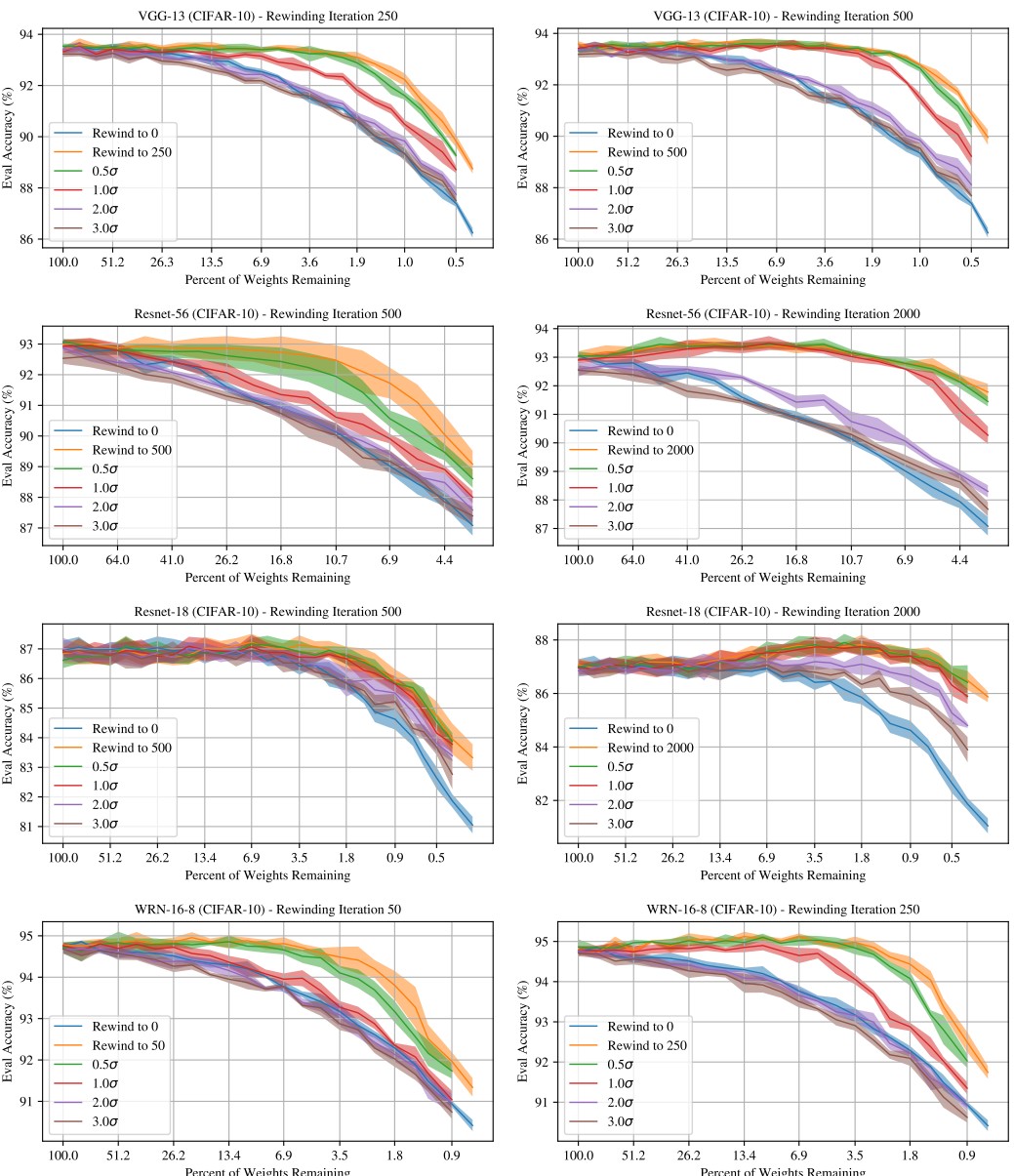

Figure A7: The effect of training an IMP-derived sub-network initialized with the weights at iteration $k$ and Gaussian noise of $n\sigma$, where $\sigma$ is the standard deviation of the initialization distribution for each layer. Accompanies Figure 7.

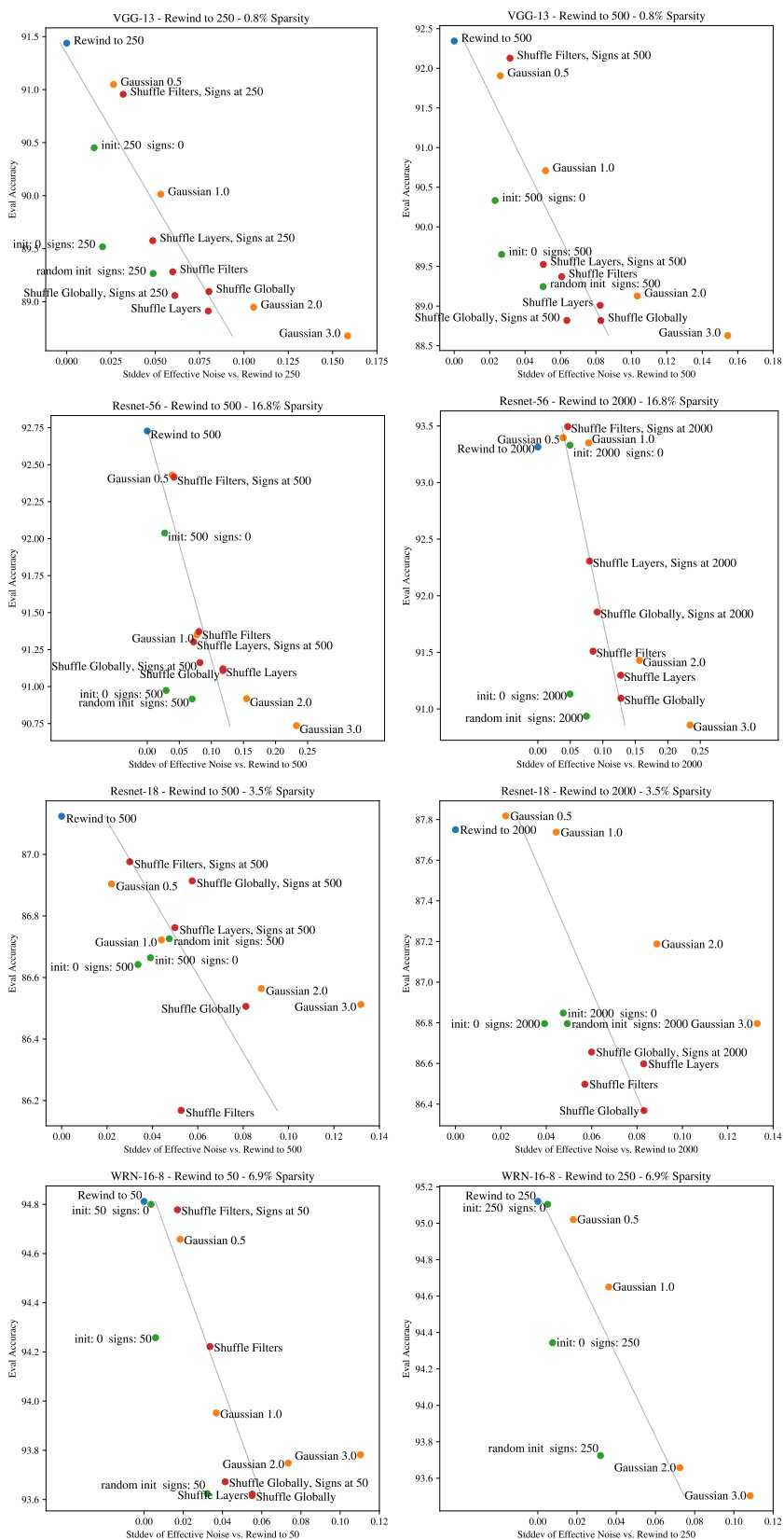

Figure A8: The effective standard deviation of each of the perturbations studied in Section 5 as a function of mean evaluation accuracy (across five seeds). Accompanies Figure 8.

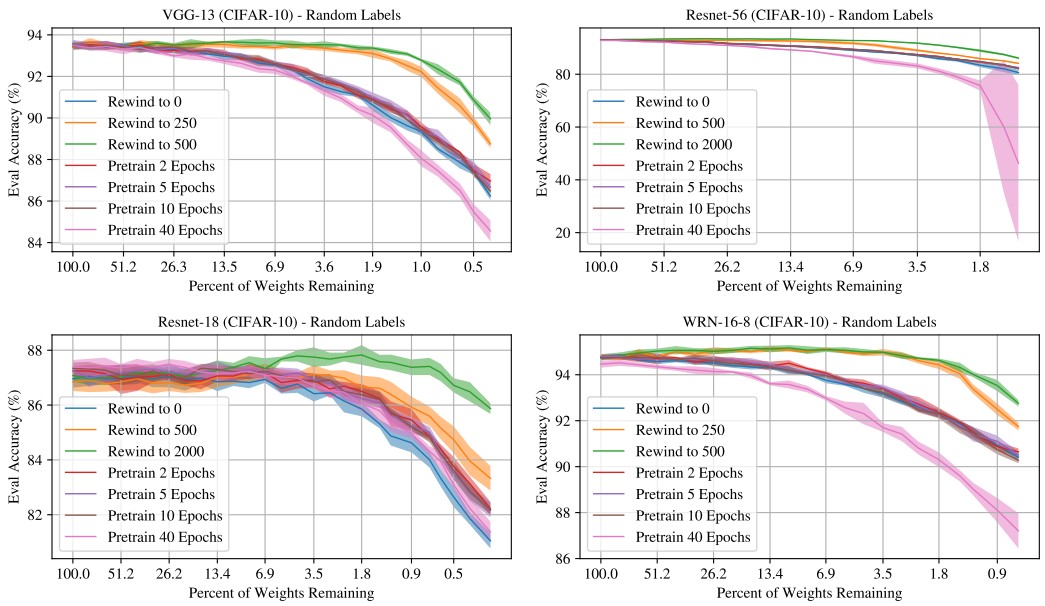

Figure A9: The effect of pre-training CIFAR-10 with random labels. Accompanies Figure 9.

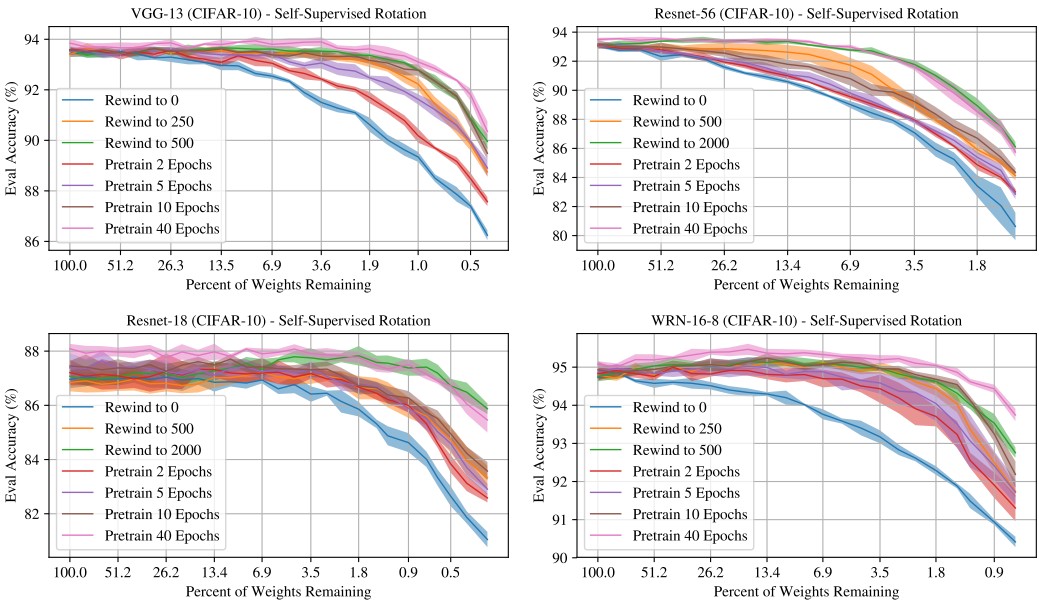

Figure A10: The effect of pre-training CIFAR-10 with self-supervised rotation. Accompanies Figure 9.

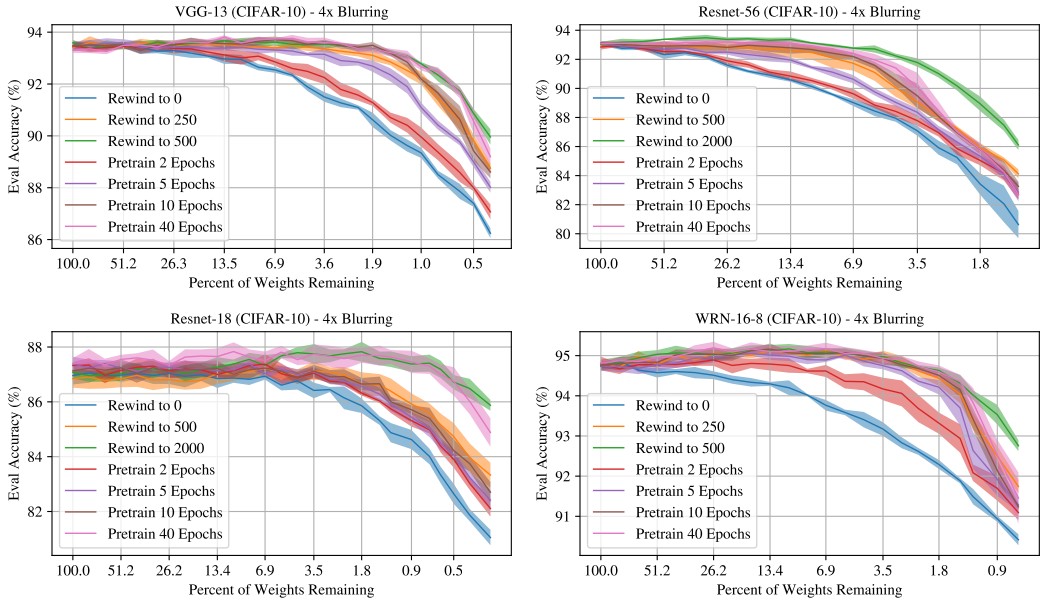

Figure A11: The effect of pre-training CIFAR-10 with 4x blurring. Accompanies Figure 9.

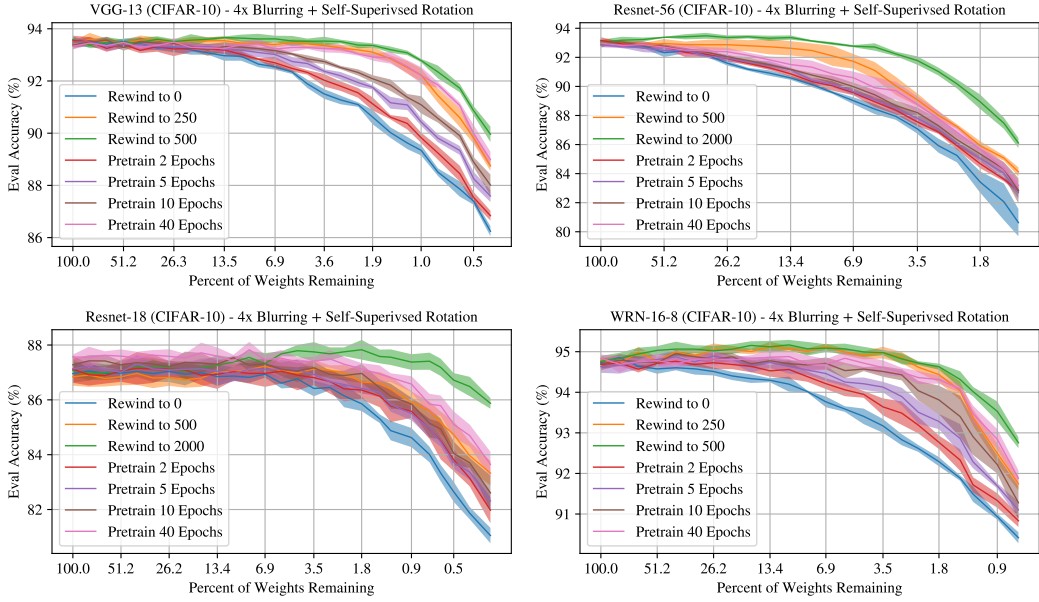

Figure A12: The effect of pre-training CIFAR-10 with 4x blurring and self-supervised rotation. Accompanies Figure 9.

