# OpenReview forum: "The Early Phase of Neural Network Training"
_ICLR.cc/2020/Conference — Accept (Poster)_

### Official Review · AnonReviewer3 · 2019-10-22
**Official Blind Review #3**

**Rating:** 6

**Review:**

Overview:

This paper is dedicated to examining the changes that networks undergo during the early phase of the network training. The author conducts extensive measurements of the network state and its updates during the early iterations of training. Based on the observations, they find that: i) deep network is not robust to reinitialization with random weights, but maintained signs; ii) the distribution is highly non-i.i.d after the early phase of network training. This is why permuting weight dramatically harms performance. iii) the changes in the supervised networks are label-agnostic. The author claims these results can play an important role in explaining the network changes in the initial critical period.

Strength Bullets:

1. The paper performs exhaustive experiments in the early phase of network training. And it has some interesting implications for lottery tickets. i.e. to some extent, then signs from the rewinding iteration are sufficient to recover the damage caused by permutation.
2. I am very like the analysis of whether weight distributions are i.i.d.. The results in the paper are aligned with my intuition that the weights in the early stage are highly dependent and they share some similarities in the distribution level. And weight in different training stages supposes in a different distribution. Networks aren't robust to these permutations. They also show that the perturbations can be roughly be approximated by adding Gaussian noise to network weights.s
3. The author offers detailed results to analysis the data-dependence of the early phase of network training. The experiment organization is complete and convincing.

Weakness Bullets:

1. Although the paper gives extensive measurements and observations about the early phase about network training, it doesn't provide useful and efficient guidance for the lottery tickets hypothesis. In other words, the observation is interesting but the novelty is arguable. How can we use these "implications" to find better tickets? I mean it should be efficient. It is not reasonable cost more the find the best weight magnitude point or sign point for a better initialization.
2. [Minor] The legend in figure 9 is missing. I guess it may be the same as the upper left. But it is still confusing and hard to read.

Recommendation:

I think this paper provides plenty of insightful observations. I still hope the author can solve the above weakness bullets. This is a weak accept.

**Experience Assessment:**

I have read many papers in this area.

**Review Assessment: Checking Correctness Of Derivations And Theory:**

I carefully checked the derivations and theory.

**Review Assessment: Checking Correctness Of Experiments:**

I carefully checked the experiments.

**Review Assessment: Thoroughness In Paper Reading:**

I read the paper at least twice and used my best judgement in assessing the paper.

---

> ### Author Response · Authors · 2019-11-08
> **Author Response to Reviewer #3**
>
> We thank the reviewer for their helpful suggestions, and have updated our paper in response to their feedback and questions. We answer specific questions below:
>
> Regarding how our results can be used to find better lottery tickets, we argue that our results do in fact provide useful and efficient guidance for the lottery ticket hypothesis. However, this guidance tells us where *not* to look, rather than how to directly improve lottery tickets. Notably, our finding that weight distributions are highly non-i.i.d. after a small number of iterations suggests that approaches to approximate rewound winning tickets through sampling from an i.i.d. distribution are unlikely to bear fruit, and that future explorations of improvements to the lottery ticket hypothesis should focus on early pruning approaches. Furthermore, our finding that the early phase of training can be approximated by training without supervision by training on rotation prediction tasks (though at the cost of longer training times) suggests that a strategy to derive winning tickets using pretraining with less supervision may be possible. We included a paragraph discussing these implications at the end of Section 7.
>
> Regarding the legend in Figure 9, we thank the reviewer for catching this error! We have added legends to all graphs in Figure 9 (and in Figures A8-A11, which contain the same data for our other networks). The legend was indeed the same for all of the graphs, but we agree that including the legend on each plot improves readability.

---

### Official Review · AnonReviewer2 · 2019-10-23
**Official Blind Review #2**

**Rating:** 8

**Review:**

This paper aims at exploring the properties of neural network training during the early phase. By some studies on the lottery ticket hypothesis, something important happens during the early phase of training so rewinding the network should go to these early phases instead of the initial phase. So, what is important during training? The paper explores this problem from four aspects through empirical studies:
1. By showing the various statistics through different training iterations, especially the gradient magnitude, the training phase is divided into three parts, and each part has different behaviors.
2. The paper explores what is more important for the early phase of training: signs of the weights or magnitude of the weights.
3. The paper explores what is more important for the early phase of training if we redistribute the weights, signs or magnitudes.
4. The paper explores how training depends on data. Giving weak information such as self-supervised information may work but giving wrong information such as random labels will hurt the performance.

This paper studies the properties of deep neural networks. Through a series of carefully designed experiments, the paper shows what is important for the weights (magnitude or signs), and what is important for the data (weak information or random information). It enables a deep understanding of neural networks and may motivate new neural network compression methods to be proposed. Generally, the paper is well-written, although some parts can be improved. I would vote for acceptance of the paper.

Some questions/suggestions to make the paper clearer:
1. It is better to have a table summarizing various results of the paper, to give readers an overall impression after going through so many detailed experimental results.
2. The results of Fig. 4 and Fig.6 can be inconsistent. In Figure 4, it says that signs are less important than magnitudes. In Figure 6, it says that signs are more important than magnitudes if shuffling filters and keep signs. Any explanation on the inconsistency?
3. There is no explanation of the “weight trace” in Figure 3.
4. It is not clear what is the difference between “Init” and “Final” in “L2 Dist” and “Cosine Similarity” in Figure 3.

Finally,  it is said in the “call for papers”, “Reviewers will be instructed to apply a higher standard to papers in excess of 8 pages”. This paper is nearly nine pages, and higher standards should be applied.

--------------------------------------
I am satisfied with the rebuttal. Since the paper is now within the 8 pages limit, I would not apply a high standard and increase my score.


**Experience Assessment:**

I have read many papers in this area.

**Review Assessment: Checking Correctness Of Derivations And Theory:**

N/A

**Review Assessment: Checking Correctness Of Experiments:**

I assessed the sensibility of the experiments.

**Review Assessment: Thoroughness In Paper Reading:**

I read the paper thoroughly.

---

> ### Author Response · Authors · 2019-11-08
> **Author Response to Reviewer #2**
>
> We thank the reviewer for the helpful suggestions to improve the clarity of our paper. We have addressed all stated concerns except for the table suggestion in the present revision. We intend to add a table summarizing results before the end of the discussion period. Below, we describe the changes we have made to the paper in detail.
>
> 1. We are working on adding a summary table of results as “Appendix A” that will summarize each of the experiments we ran, each of the dependent variables, and the key takeaways. We agree that this will be very helpful for readers. We are still determining how exactly to structure this table as there are many experiments, and we will update the paper again once it is ready.
>
> 2. While we understand why Figures 4 and 6 may seem at odds with one another, their relationship to one another is more nuanced based on which rewinding iteration is analyzed. We clarify the relationship below, and we have updated the second paragraph of Section 5.2 to make clear the differences.
>
> For rewinding iteration 500, the magnitudes from iteration 500 are more important than the signs. In Figure 4 (left), having the magnitudes alone leads to improved performance (purple line), while having the signs alone leads to bad performance (green and red lines). In every experiment in Figure 6 (left), the magnitudes have been shuffled and performance correspondingly drops, even when fixing the signs. The sole exception is “Shuffle Filters,” which, as Figure 8 shows, is a relatively weak perturbation.
>
> For rewinding iteration 2000, the magnitudes from iteration 2000 are sufficient for improved performance, but the signs only lead to improved performance in limited settings. Namely, we only see this behavior when the signs are paired with the original initialization (red line in Figure 4 right) or with magnitudes from iteration 2000 (red and green lines in Figure 6 right). When paired with a random initialization (green line in Figure 4 right), performance is poor.
>
> Our interpretation is that the signs at iteration 2000 only lead to improved performance when the magnitudes are similar to their values in 2000, whether distributionally (the shuffle experiments) or in the values themselves (using the original initialization, whose values will be more similar than random values).
>
> 3. The “weight trace” graph plots the values of ten randomly-selected weights over the first ten epochs of training. Some weights remain roughly constant while others vary wildly; since there was no general behavior, we did not analyze this graph in the text. We have updated the caption of Figure 3 to clarify the data that is presented in this graph.
>
> 4. The “init” and “final” referred to the L2 and cosine distance from the weights at each iteration to the initialization and to the final weights at the end of training. In other words, these metrics show how far the network has moved from its starting point (“init”) and how far it still has to move to reach its converged point (“final”). We have updated the caption of Figure 3 to clarify the meaning of these terms.
>
> Finally, we have reduced the length of our paper so that it fits within 8 pages by moving the  experiments where we pretrain already-sparse networks from Section 6.4 to the appendix.

---

### Official Review · AnonReviewer1 · 2019-10-24
**Official Blind Review #1**

**Rating:** 3

**Review:**

Pros
+ This work provided a good summary of observations and network properties that worth studying during the early stage of network’s training.
+ The authors conducted extensive and detailed experiments to study the statistics of weights and their gradients. Ablation studies also considered network’s accuracy under perturbation of weight signs, weight shuffling, and different standard deviations.
+ The authors also verified the effectiveness of weak labels used in self-supervised learning.

Cons
- The work itself did not propose any new network properties or any new metric to measure. Most experiments are designed for previous observations and mostly for verification purpose. I am concern about the core motivation of this work, like to identify or solve any new problems, in addition to experimentally verify the observations during network’s training.
- The conclusion in this work is still very empirical: it remains uncertain whether in other vision tasks and with more complex networks (e.g. multi-branch network) these conclusions would hold.

**Experience Assessment:**

I have published one or two papers in this area.

**Review Assessment: Checking Correctness Of Derivations And Theory:**

I carefully checked the derivations and theory.

**Review Assessment: Checking Correctness Of Experiments:**

I carefully checked the experiments.

**Review Assessment: Thoroughness In Paper Reading:**

I read the paper at least twice and used my best judgement in assessing the paper.

---

> ### Author Response · Authors · 2019-11-08
> **Author Response to Reviewer #1**
>
> We thank the reviewer for their comments. Our response to their concerns is below:
>
> While the reviewer is correct that our work does not propose any strictly new metrics, we would like to emphasize that discovering new metrics was not the goal of this work. Rather, our aim was to rigorously and exhaustively measure a number of network properties as they change over the initial portion of training in order to better understand why this period of learning is so critical and to elucidate the precise factors responsible for these changes. This is best done using existing metrics that are already meaningful to the reader.
>
> While novel algorithms and metrics are of course important, careful empirical work aiming to better understand these metrics and algorithms are equally important. This style of work is increasingly being recognized and sought by the community, as evidenced by several recent workshops and talks emphasizing rigorous empirical studies [1, 2, 3].
>
> Moreover, we note that we have developed an entirely new scientific application for the lottery ticket framework as a means to investigate the early phase of training, and our experiments have several clear implications for a number of downstream investigations, including better pruning strategies and initializations. First, we found that, in contrast to previous work on smaller networks [4], reinitializing networks with random weights but maintained signs is not in fact sufficient for more commonly used deeper networks, such as ResNets. Second, we found that even after only a small number of iterations, the weight distributions are highly non-i.i.d., suggesting that approximating such a distribution to sample from for initialization, as was implied by initial work into the lottery ticket hypothesis, is unlikely to succeed (at least without jointly modeling the weights). Finally, we demonstrated that the changes which take place over the initial course of training *can* in fact be approximated by unsupervised pretraining, which suggests several avenues for improving training of compressed networks. We believe that all of these conclusions will likely be important for future investigations into ways to improve training. Reviewer 2 agreed that these conclusions will likely have impact on future network improvements, stating that “...[our study] may motivate new neural network compression methods to be proposed.”
>
> While the reviewer is correct to note that our experiments were only performed on CIFAR-10, primarily due to compute limitations (each line on each plot represents approximately 100 individual model trainings, making equally rigorous experiments on larger datasets prohibitively expensive), we emphasize that our conclusions were robust across five distinct network architectures: ResNet-20, ResNet-56, ResNet-18, Wide-ResNet-16-8, and VGG-13. While we were only able to include ResNet-20 in the main text due to space limitations, we have included plots for the other four networks for all experiments in the appendix section C. We were glad to see all three reviewers recognize the breadth of experiments, calling them “extensive and detailed” (R1), “carefully designed” (R2), and “exhaustive” (R3). Therefore, while we agree with the reviewer that further work for other task types would be interesting, we believe it is beyond the scope of the present work.
>
> [1] Identifying and Understanding Deep Learning Phenomena, ICML 2019 workshop, http://deep-phenomena.org/
>
> [2] Science meets Engineering of Deep Learning, NeurIPS 2019 workshop, https://sites.google.com/view/sedl-neurips-2019/main
>
> [3] Ali Rahimi, Test of Time award talk, NeurIPS 2017, https://www.youtube.com/watch?v=Qi1Yry33TQE
>
> [4] Hattie Zhou, Janice Lan, Rosanne Liu, and Jason Yosinski. Deconstructing lottery tickets: Zeros, signs, and the supermask. NeurIPS, 2019.

---

### Decision · Program_Chairs · 2019-12-19

**Decision:**

Accept (Poster)

**Comment:**

This paper studies numerous ways in which the statistics of network weights evolve during network training.  Reviewers are not entirely sure what conclusions to make from these studies, and training dynamics can be strongly impacted by arbitrary choices made in the training process.  Despite these issues, the reviewers think the observed results are interesting enough to clear the bar for publication.